# Roles of Non-Coding RNAs in Response to Nitrogen Availability in Plants

**DOI:** 10.3390/ijms21228508

**Published:** 2020-11-12

**Authors:** Makiha Fukuda, Toru Fujiwara, Sho Nishida

**Affiliations:** 1Institute for Systems Genetics and Department of Biochemistry and Molecular Pharmacology, New York University Langone Health, New York, NY 10016, USA; makiha.fukuda@nyulangone.org; 2Department of Applied Biological Chemistry, Graduate School of Agricultural and Life Sciences, The University of Tokyo, 1-1-1 Yayoi, Bunkyo-ku, Tokyo 113-8657, Japan; atorufu@mail.ecc.u-tokyo.ac.jp; 3Department of Bioresource Science, Faculty of Agriculture, Saga University, 1 Honjo-machi, Saga 840-8502, Japan

**Keywords:** long non-coding RNA, microRNA, nitrogen, plant nutrition, small interfering RNA

## Abstract

Nitrogen (N) is an essential nutrient for plant growth and development; therefore, N deficiency is a major limiting factor in crop production. Plants have evolved mechanisms to cope with N deficiency, and the role of protein-coding genes in these mechanisms has been well studied. In the last decades, regulatory non-coding RNAs (ncRNAs), such as microRNAs (miRNAs), small interfering RNAs (siRNAs), and long ncRNAs (lncRNAs), have emerged as important regulators of gene expression in diverse biological processes. Recent advances in technologies for transcriptome analysis have enabled identification of N-responsive ncRNAs on a genome-wide scale. Characterization of these ncRNAs is expected to improve our understanding of the gene regulatory mechanisms of N response. In this review, we highlight recent progress in identification and characterization of N-responsive ncRNAs in *Arabidopsis thaliana* and several other plant species including maize, rice, and *Populus*.

## 1. Introduction

Plants require 17 essential elements (C, H, O, N, P, K, Mg, Ca, S, Cl, Fe, B, Mn, Zn, Cu, Ni, and Mo) to complete their life cycle [1]. Nitrogen (N) is an essential macronutrient, and 1–5% of total plant dry matter consists of N. The major sources of N are nitrate and ammonium, which are absorbed from the soil via nitrate transporters (NRTs) and ammonium transporters (AMTs), respectively [1]. Absorbed N is assimilated and used as a source of metabolites critical for plant survival, such as amino acids, nucleic acids, chlorophyll, coenzymes, phytohormones, and secondary metabolites [1]. It is therefore a major limiting factor for plant growth and crop production [2]. Availability of N is affected by soil properties, such as pH and moisture [3], and plants have evolved sophisticated mechanisms to adapt to variable N availability.

Plants optimize the activities of N absorption and assimilation in response to N availability in soils by regulating expression of genes involved in these processes. In *Arabidopsis thaliana*, expression of genes encoding high-affinity NRTs (*AtNRT2.1*, *AtNRT2.4*, and *AtNRT2.5*) is induced in roots under N deficiency [4]. In contrast, under N sufficiency, *AtNRT2* genes are down-regulated, and nitrate absorption is switched to low-affinity mode via the AtNRT1 family [5,6]. Genes encoding nitrate reductase (NR) and nitrite reductase (NiR) are induced by nitrate supply [7,8] and facilitate its assimilation, whereas these genes are down-regulated under N deficiency, which is likely in order to limit nitrate assimilation activity under low-nitrate supply. In *A. thaliana*, several transcription factors are responsible for transcriptional regulation of these genes [9]. NIN-like protein (NLP) transcription factors are responsible for regulation of *NR* and *NiR* expression in response to nitrate supply [10,11,12]. NITRATE-INDUCIBLE GARP-TYPE TRANSCRIPTIONAL REPRESSOR 1 proteins (NIGT1s) directly repress *AtNRT2* and suppress high-affinity nitrate transport under N sufficiency [13,14]. Expression of *NIGT1*s is negatively autoregulated and is also regulated by NLPs. Integration of the *NLP*–*NIGT1* cascade and *NIGT1* autoregulation seems to orchestrate the nitrate response [13].

Modification of root architecture is also important for plant adaptation to changing N availability [15,16,17,18] and has been well studied in *A. thaliana*. Lateral root (LR) growth is stimulated under moderate N deficiency (275 or 550 µM in media [19]), enabling plants to explore available N. Stimulation of LR growth under moderate N deficiency is thought to be caused by induction of *TRYPTOPHAN AMINOTRANSFERASE-RELATED 2* (*TAR2*), an auxin biosynthetic gene involved in development of LR primordia [20]. By contrast, under severe N deficiency, LR growth is repressed [19,21,22], which likely restricts investment in extending roots into N-poor environments [22]. Inhibition of LR growth under severe N deficiency is mediated by CLAVATA3/ESR-related (CLE) signaling peptides and their receptor protein, CLAVATA1 (CLV1) [22,23]. *CLE* genes (*CLE1*, *CLE3*, *CLE4*, and *CLE7*) are induced in response to N deficiency in root cells, and accumulated CLE peptides interact with CLV1, resulting in repression of genes involved in LR development [22,23]. Repression of LR growth under severe N deficiency is also caused by NRT1.1: NRT1.1 transports not only nitrate but also auxin and facilitates shootward auxin movement in LRs, resulting in reduced auxin concentration in LR tips and repressed LR growth [21].

Despite these advances in research on plant response to N availability, we are still far from understanding the molecular mechanisms underlying complex spatiotemporal regulation of hundreds of genes in response to N availability. Over the last decade, a large number of non-coding RNAs (ncRNAs) have been discovered that are distinct from well-known housekeeping RNAs such as ribosomal RNA, transfer RNA, small nuclear RNA, and small nucleolar RNA [24]. The proportion of functional ncRNAs remains unclear, but dozens of ncRNAs regulate gene expression at transcriptional and post-transcriptional levels [25,26]. In plants, expression of many ncRNAs responds to various environmental stimuli [27,28,29,30,31,32,33,34,35,36], indicating their roles in tolerance to environmental stresses such as drought, salinity, heat, cold, and nutrient deficiency [37,38,39]. Advances in RNA sequencing (RNA-Seq) technology have enabled identification of ncRNAs with expression affected by N availability (N-responsive ncRNAs) in various plants including *A. thaliana*, *Populus* (*Populus tomentosa*), rice (*Oryza sativa*), and maize (*Zea mays*) [30,40,41,42,43]. Very recently, the role of N-responsive ncRNAs in plant response to N availability has been demonstrated. In this review, we summarize identification and characterization of N-responsive ncRNAs and discuss their importance in plant response to N availability.

## 2. Classification and Functions of Plant ncRNAs

### 2.1. Classification of ncRNAs

ncRNAs can be either housekeeping or regulatory. The former group has a housekeeping role in transcription and translation, whereas the latter regulates genes [44]. Regulatory ncRNAs are further classified into small ncRNAs (microRNAs, miRNAs, small interfering RNAs, and siRNAs), and long ncRNAs (lncRNAs, typically > 200 nt and lacking the potential to code proteins > 100 amino acid residues) [39].

### 2.2. Small Regulatory ncRNAs

The major difference between miRNAs and siRNAs is their biogenesis: miRNAs are processed from transcripts that can form local RNA hairpin precursor structures, whereas siRNAs are processed from long bimolecular RNA duplexes or extended hairpins [45]. Initially, miRNAs are transcribed from miRNA host genes (*MIR* genes), predominantly by RNA polymerase II, as precursors called primary miRNAs (pri-miRNAs). Mammalian miRNA host genes often encode multiple miRNAs and are frequently nested within protein-coding genes, whereas their plant counterparts are independent transcription units with a single miRNA [46]. Pri-miRNAs form intra-molecular hairpin-like structures and are processed into mature miRNAs (typically 20–22 nt) by DICER-LIKE (DCL) RNase III endonucleases, mainly DCL1 [25,47,48,49]. Mature miRNAs are loaded into ARGONAUTE 1 (AGO1) to form the RNA-induced silencing complex (RISC) and then guide the RISC to the target mRNA through sequence complementarity and trigger gene silencing [50,51,52].

Depending on the organism and the cellular context, siRNA precursors, double-stranded RNAs (dsRNAs), may arise from bidirectional transcription (sense and antisense transcripts) of a chromosomal locus or from unidirectional transcription followed by RNA-dependent RNA polymerase activity [53]. Endogenous plant siRNAs can be grouped into three classes [54]. A major class is generated from heterochromatic regions (such as centromeres, transposons, and other repetitive elements); these precursors are transcribed predominantly by the plant-specific polymerase Pol IV, followed by processing into so-called heterochromatic siRNAs (hc-siRNAs; 24 nt) [55,56]. These hc-siRNAs induce DNA methylation and histone modifications through RNA-directed DNA methylation [57,58]; their mode of action is consistent with the idea that RNA interference (RNAi) functions in silencing transposon expression and propagation [59]. Another class of siRNAs, so-called secondary siRNAs, require an initial small-RNA-directed cleavage of the primary transcript to trigger dsRNA synthesis [60,61,62]. Among siRNAs of this class, *trans*-acting siRNAs (tasiRNAs) are best studied and, unlike many other siRNAs, act in *trans* to direct post-transcriptional repression of their mRNA targets [60]. Precursors of the third siRNA class are formed from mRNAs encoded by natural *cis*-antisense gene pairs, which are transcribed from opposite DNA strands of the same genomic loci (*cis*-NATs) [63]. Similar to the case of miRNAs, siRNA precursors are processed by DCLs into small RNAs and then loaded into AGO proteins to form the RISC [64,65,66,67], which silences a target gene.

In addition to their role in transcriptional silencing, intercellular mobility of small RNAs is another important aspect to be considered. Small RNAs can move either locally (cell-to-cell) or systemically (organ-to-organ) and play a role as mobile silencing signals. Several nutrient-responsive miRNAs are known to be accumulated in the phloem during starvation, indicating that those miRNAs are systemically translocated in the plant to amplify silencing signals (reviewed in [68]). Such a non-cell autonomous effect of small RNAs likely enables rapid adaptation to changing environments.

### 2.3. Long Regulatory ncRNAs

LncRNAs are transcribed from intergenic, intronic, and coding regions, and regulate gene expression [26,69]. Some lncRNAs are precursors for siRNAs or miRNAs, whereas others function without being processed into small RNAs [26]. Some lncRNAs function as molecular decoys that sequester proteins or small RNAs from their target RNA as in the case of *IPS1* [70] and *ASCO* [71]. LncRNAs can also associate with chromatin and control gene transcription. HIDDEN TREASURE 1 (HID1) interacts with the promoter region of the transcription factor gene *PIF3* and represses its expression to regulate photomorphogenesis [72]. The well-known flowering-related lncRNAs *COLDAIR*, *COLDWRAP*, and *COOLAIR* are transcribed from *FLOWERING LOCUS C* (*FLC*) and interact with Polycomb proteins, thereby controlling epigenetic silencing of *FLC* mediated by histone modifications [73,74,75].

### 2.4. Plant ncRNA Databases

Although mammalian ncRNAs remain the best studied, with the help of recent advances in sequencing technologies, identification of plant ncRNAs has almost caught up with the mammalian field. Several databases for plant ncRNAs have been established [24]; the *A. thaliana* small RNA project (ASRP) database contains 218,585 unique small RNA sequences [76], and GREENC contains more than 120,000 lncRNAs from 37 plant species [77]. Many plant ncRNAs are expressed in response to abiotic stresses including drought, temperature, salinity, and nutrient deprivation [27,28,30,78]. Recently, a database of experimentally validated stress-responsive ncRNAs, named PncStress, has been established, which contains 4227 entries including lncRNAs and miRNAs from 114 plant species and covers 91 abiotic stresses and 48 biotic stresses [79].

## 3. N-Responsive ncRNAs in *A. thaliana*

### 3.1. miRNAs Involved in N Nutrition

Among N-responsive ncRNAs in *A. thaliana*, miRNAs have been relatively well studied. Several miRNAs have been characterized as associated with N stresses [80,81,82,83]. The first report on the role of a miRNA in N response was identification of miR167 and its target gene *AUXIN RESPONSE FACTOR 8* (*ARF8*); nitrate influx suppresses expression of miR167, allowing ARF8 to accumulate in the root pericycle cells and to initiate LR outgrowth [84]. N-starvation-responsive miRNAs miR160 and miR171 [81] regulate root system architecture by suppressing expression of their target transcription factor genes *ARF10/16/17* and *SCARECROW-LIKE PROTEIN 6* (*SCL6*), respectively [85,86]. miR393 and its target *AUXIN-SIGNALING F-BOX PROTEIN 3* (*AFB3*) also control the responses of root system architecture to both external and internal N [87,88]. Expression of *AFB3* is induced by nitrate and promotes LR formation, whereas miR393 is induced by N metabolites, thus allowing plants to respond appropriately to N conditions [87]. miR169 down-regulates N uptake by inhibiting expression of its target gene, *NUCLEAR FACTOR Y SUBUNIT A5* (*NFYA5*), a positive regulator of nitrate transporter genes *AtNRT2.1* and *AtNRT1.1* [89]. miR826 and miR5090 regulate N metabolism through their common target gene *ALKENYL HYDROXALKYL PRODUCING 2* (*AOP2*), which encodes 2-oxoglutarate-dependent dioxygenase involved in synthesis of N-containing metabolites called glucosinolates [90]. Under N starvation, expression of *AOP2* is post-transcriptionally repressed by miR826 and miR5090, decreasing glucosinolate synthesis and thus enabling plants to distribute N to metabolites required for survival [91]. Interestingly, both miRNAs may have evolved by inverted duplication of the genomic region containing *AOP2*, implying coevolution of miR826, miR5090, and *AOP2* in adaptation to N deprivation [91,92].

### 3.2. Genome-Wide Identification of N-Responsive miRNAs

N-responsive miRNAs have been identified by isolation of small RNAs from total RNA by size fractionation followed by sequencing. Initially, a genome-wide search of nitrate-induced miRNAs using 454-sequencing technology identified miR393 [93]. Subsequently, using Illumina sequencing, Liang et al. [81] reported expression profiles of 177 miRNAs under N-sufficient and deficient conditions; some of these miRNAs may be involved in root system development in response to N [81]. The authors also identified miRNAs responsive to carbon and sulfate, which may help to explain crosstalk between different nutrient responses [94]. Hundreds of N-responsive miRNAs have been identified, but their roles remain largely uncharacterized.

### 3.3. T5120 and TAS3: lncRNAs Involved in N Response

Recently, *T5120* and *TAS3* have been reported to be involved in N response [30,40]. Expression of *T5120* was significantly induced by 2h treatment of *A. thaliana* roots with nitrate [40]. Induction was mediated by direct binding of transcription factor NLP7 to the promoter region in a nitrate-dependent manner. NLP7 acts on nitrate-responsive genes, such as *NRT2.1*, *NIR*, *NRT2.2*, and *NIA*, and orchestrates early response to nitrate [95]. In transgenic *A. thaliana*, overexpression of *T5120* significantly increases nitrate reductase activity, amino acid content, seedling biomass, and expression of nitrate-assimilatory genes including *NIA1*, *NIA2*, *NIR*, and *GLN1.1*, whereas nitrate uptake is not affected [40]. Interestingly, *T5120* gene is positioned adjacent to *NIA1* on the chromosome and might regulate its expression by affecting chromatin structure or modification [40], as in the case of lncRNAs *APOLO*, and *COOLAIR* [73,96]. Further investigation of regulatory mechanisms behind *T5120*-mediated control of N assimilation is required for *T5120* engineering to develop crops with improved N use efficiency.

*TAS3* is a precursor of tasiRNAs called tasiARFs, which target *ARF2*, *ARF3*, and *ARF4* [97]. *TAS3* transcripts are bound and cleaved by miR390, an evolutionarily conserved miRNA, which triggers tasiARF production [60]. The miR390/*TAS3*/*ARF* pathway is part of the auxin-mediated molecular network that orchestrates LR formation in *A. thaliana* [98,99]. Recently, it was found by RNA-Seq analysis that expression of *TAS3* is significantly reduced under severe N deficiency [30]. Because *TAS3* positively regulates LR growth [98], we hypothesized that *TAS3* down-regulation in response to N deficiency suppresses LR outgrowth by increasing expression of *ARFs*. Consistent with this hypothesis, expression of *ARF2–4*, especially *ARF4*, is up-regulated by N deficiency, whereas *TAS3* overexpression down-regulates *ARF4* and significantly increases LR length, even under severe N deficiency [30]. Interestingly, we found that *TAS3* decreases not only expression of *ARFs* but also that of *NRT2.4* [100]. Our computational analysis suggests an interaction between *TAS3*-derived small RNA and the *NRT2.4* transcript [30]. We confirmed *TAS3*-dependent cleavage of the *NRT2.4* transcript by rapid amplification of cDNA ends [30]. These observations indicate the role of *TAS3* as a source of multiple tasiRNAs that regulate multiple signaling pathways depending on N availability. Inhibition of LR growth in response to severe N deficiency has been explained by NRT1.1-mediated auxin transport [21] and CLE–CLV1 peptide-receptor signaling module [22,23]. We propose an additional regulatory mechanism involving *TAS3*-derived tasiARFs (Figure 1). In parallel, *TAS3* might also regulate N uptake cooperative with the NIGT1.1/NRT2.4 transcriptional cascade [14,100] (Figure 1).

### 3.4. Genome-Wide Identification of N-Responsive lncRNAs

RNA-Seq data have provided a catalog of N-responsive lncRNAs. Liu et al. [40] analyzed the root transcriptome of 7-day-old *A. thaliana* roots treated with 10 mM KNO_3_ for 2 h and identified 7 up-regulated and 1 down-regulated lncRNAs among a total of 695 lncRNAs (|log_2_ fold change| ≥ 1 vs. untreated controls) [40]. We have analyzed the transcriptome of *A. thaliana* roots exposed to 12 different nutrient starvation conditions (-N, -K, -Ca, -Mg, -P, -S, -B, -Fe, -Mn, -Zn, -Cu, or -Mo) for 4 days and identified 60 lncRNAs differentially expressed (|log_2_ fold change| ≥ 1) under at least one nutrient deficiency in comparison with nutrient-sufficient controls [30,101]. Among them, 6 lncRNAs were up-regulated and 15 were down-regulated in response to N deficiency. Among these 21 lncRNAs, *AT1G11185*, *AT1G67105*, and *TAS3* have been also reported to be N responsive by Liu et al. [40]. For further characterization of lncRNAs, we have developed a bioinformatics pipeline to predict molecular interactions between lncRNAs and mRNAs, and have identified hundreds of lncRNA–mRNA pairs with low interaction energy (high interaction potential) and high correlation of expression patterns. Among these pairs, *AT1G67105* was predicted to bind the mRNA of an ammonium transporter gene *AMT1;2* [30,102]. Under N deficiency, *AT1G67105* is down-regulated, whereas *AMT1;2* is up-regulated, indicating the possible role of *AT1G67105* in regulation of ammonium uptake by controlling expression of *AMT1;2* [30]. Our study provided tens of candidate lncRNAs that function in low-nutrient responses and proposed an approach for predicting their target genes. This approach would help to explore how those lncRNAs contribute to gene regulation under nutrient deficiencies.

## 4. N-Responsive ncRNAs in Other Plant Species

### 4.1. Maize ncRNAs

Genome-wide microarray-based analysis identified 14 differentially expressed miRNAs in maize (miR160, miR164, miR167, miR168, miR169, miR172, miR319, miR395, miR397, miR398, miR399, miR408, miR528, and miR827) under chronic (15 days) or transient (2 h) N deficiency [103]. They were categorized into miRNAs targeting (i) transcription factors (e.g., miR167–*ARF8* [104]), (ii) genes involved in energy metabolism (e.g., miR398–*COX* [105]), and (iii) genes involved in miRNA regulation (e.g., miR168–*AGO1* [106,107]). Using in situ hybridization, Trevisan et al. [108,109] characterized spatial distribution of N-responsive miRNAs miR528a/b, miR169i/j/k, miR166j/k/n, and miR408/b under different N conditions [108]. Deep sequencing technologies have accelerated identification of N-responsive miRNAs. Of a total of 85 miRNAs newly identified by small-RNA sequencing, 25 showed > two-fold relative change in response to low N [110,111]. Another study reported that 106 miRNAs were differentially expressed between the control and N-deficient groups and 103 between N-deficient and N-resupply groups [112].

Of 7245 lncRNAs identified by RNA-Seq analysis in maize leaves under N sufficiency or N deficiency, 637 were responsive to low N [113]. Coexpression analyses suggested that most of those lncRNAs are involved in energy metabolic pathways related to NADH dehydrogenase activity, oxidative phosphorylation, and the N compound metabolic process; however, the exact function of each lncRNA remains uncharacterized.

### 4.2. Rice ncRNAs

Expression of miRNAs in rice roots exposed to different N sources and concentrations has been comprehensively analyzed by small RNA-Seq [114]. In N-starved rice shoots and roots, a combination of different sequencing datasets (strand-specific RNA-Seq, small RNA-Seq, poly(A)-primed sequencing (2P-Seq), and degradome sequencing) has provided comparative expression profiles of protein-coding genes, lncRNAs, and miRNAs along with their putative target genes [42]. This study identified 918 differentially expressed lncRNAs and 91 target genes of 40 miRNAs [42]. The authors have proposed a new model for small RNA generation from two *cis*-NAT pairs, *AMT1.1*–*cis*-NAT*_AMT1.1_* and *AMT1.2*–*cis*-NAT*_AMT1.2_*. In general, mRNA–*cis*-NAT pairs whose partial or entire sequences are complementary to other transcripts form dsRNAs that can be cleaved by a Dicer endonuclease to produce small RNAs [63,115]. However, the authors have shown that small RNAs are generated from *AMT1* mRNA itself rather than from dsRNAs, indicating a different mechanism of small RNA biogenesis [42]. As *cis*-NATs are transcribed from the *AMT* gene loci of other species such as *A. thaliana*, barley (*Hordeum vulgare*), and maize, *cis*-NAT*_AMT_* might have an evolutionally conserved role in N response.

### 4.3. Populus ncRNAs

*Populus* is a fast-growing tree of high economic and environmental value, especially in timber production and forestry. Expression profiling of ncRNAs in *P. tomentosa* plantlets under N deficiency has recently been performed [41,116,117]. Using small RNA-Seq and degradome sequencing, 60 N-responsive miRNAs were identified including 39 conserved, 13 non-conserved, and 8 novel miRNAs, along with their potential target genes [116,117]. Chen et al. [41] identified 126 differentially expressed lncRNAs in N-starved plantlets and classified them into pri-miRNAs, miRNA targets, and antisense lncRNAs. These studies lay the foundation for further studies on molecular mechanisms of N response in *Populus* and will help to engineer woody plants with improved N-use efficiency. There is no report available on genome-wide identification of N-responsive ncRNAs in other tree species at this time.

### 4.4. ncRNAs in Other Crops

Further to the plant species described above, N-responsive miRNAs have also been identified from dicot crop species including soybean (*Glycine max*) [118], potato (Indian potato variety *Kufri Jyoti*) [119], tea plant (*Camellia sinensis*) [120], and from other monocot species such as barley (*Hordeum vulgare*) [121] and wheat (*Triticum aestivum* and *Triticum turgidum*) [122,123,124,125], while N-responsive lncRNAs have yet to be systematically explored.

## 5. Evolution of ncRNAs

It is now obvious that the plant genome produces tens of thousands of lncRNAs [126]. The sequences of lncRNAs are less conserved than those of mRNAs [127,128,129], and lncRNAs are often transcribed in a strict stress-specific manner [37], implying that many of them play an ancillary rather than essential role. This raises the question of why the genome contains so many ncRNA genes, comparable to the number of protein-coding genes. The relatively low essentiality of ncRNAs could be explained by their origins. Investigations of the origin of lncRNAs have focused on possible contributions of protein-coding genes and transposons. Transposable elements (TEs) are a major force shaping the lncRNA repertoire through their capacity to introduce regulatory sequences essential for transcription, such as transcription initiation, splice, and polyadenylation sites, upon chromosomal insertion both in mammals and plants [130,131,132]. In plants, most of non-TE lncRNAs (ranging from 50% to 70%, depending on species) are predicted to originate from divergent transcription at promoters of active protein-coding genes [133]. Another large fraction (20.2% on average) of non-TE lncRNAs are located within the 2-kb proximal upstream regions of pseudogenes [133], indicating that pseudogenes also contribute to the makeup, evolutionary origins, and regulation of lncRNAs (Figure 2). In either case, lncRNA loci have originated through pre-existing gene regulatory sequences; therefore, they are likely to have evolved later than most protein-coding genes. lncRNAs seem to have evolved to modulate pre-existing gene networks and, for example, improve adaptation to environmental stresses for organism survival. Such adaptive evolution is advantageous, especially for plants, which are sessile. Identification of miRNAs from various plant species revealed that plant *MIR* genes have originated mainly by duplication of pre-existing *MIR* genes or protein-coding genes [134]. In addition, only a few miRNA families are conserved [135], suggesting that the majority of miRNAs have also emerged later than protein-coding genes. Plants seem to have evolved and diversified *MIR* genes in a lineage-specific manner, which may have allowed organisms to develop more complex gene regulatory networks to adapt to various stress conditions. From a practical perspective, ncRNAs are attractive targets of genome engineering to enhance the stress tolerance of crops without disrupting any essential genes.

## 6. Conclusions and Perspectives

N response in plants has been extensively studied from multiple perspectives such as morphology, development, fertility, metabolites, hormonal signaling, and gene expression. However, how the expression of N-responsive genes is differentially and appropriately regulated depending on N conditions remains to be further investigated. ncRNAs are important regulators of gene expression, and molecular and functional characterization of N-responsive ncRNAs will extend our understanding of gene regulatory mechanisms in N response. Although a large number of N-responsive ncRNAs have been identified in several different plant species over the last decades, dissecting their role and regulation remains a major challenge. Further studies on ncRNAs that provide experimental validation of their biological function are of great importance in the coming decades. Moreover, in silico approaches to predict ncRNA functions using multiple sources of different high-throughput data are necessary to accelerate ncRNA characterization [136]. Engineering of N-responsive ncRNAs could facilitate development of crops with improved N-use efficiency and/or low-N tolerance. For example, overexpression of miR169o, which is down-regulated by N deficiency, significantly improves N use efficiency in rice [137]. This approach could increase crop yields and promote sustainable agriculture while reducing use of N fertilizers and environmental pollution [138]. However, ncRNA overexpression using a constitutive promoter may be prone to cause undesirable effects, such as reduced resistance to bacterial infection as is the case with miR169o overexpression [137]. Advanced strategies for ncRNA engineering that allow more precise and specific control of expression and/or function of ncRNA will be required.

## Figures and Tables

**Figure 1 ijms-21-08508-f001:**
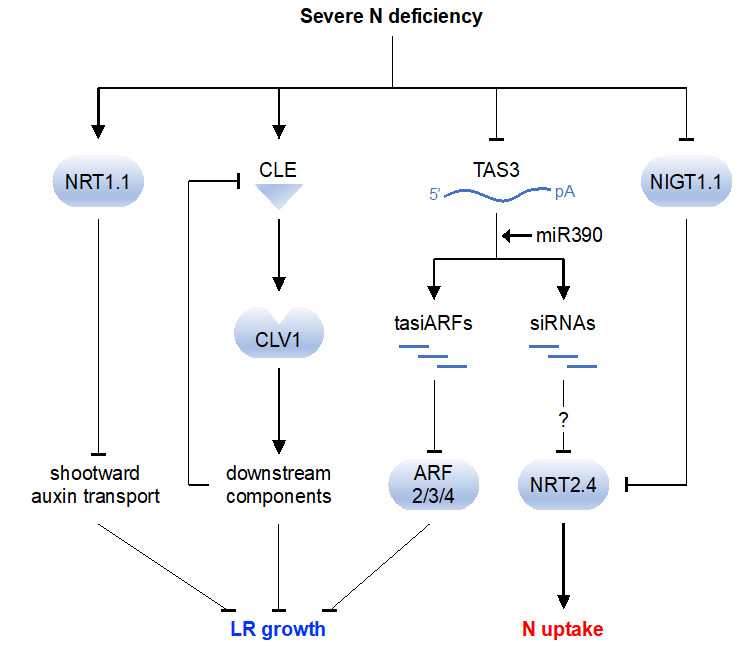
Signaling pathways modulating lateral root (LR) growth and N uptake under severe N deficiency in *A. thaliana*. Under low-N conditions, expression of *TAS3* is suppressed; therefore, production of tasiARFs is reduced [30]. As a result, expression of *ARF2/3/4* is derepressed and LR growth is inhibited [98]. Inhibition of LR growth is also controlled by NRT1.1-mediated auxin transport [21] and the CLE–CLAVATA1 (CLV1) peptide–receptor signaling module [22]. On the other hand, *TAS3* down-regulates expression of *NRT2.4* by inducing cleavage of *NRT2.4* mRNA [30]. Considering that *NRT2.4* is transcriptionally suppressed by NIGT1.1 [14], *TAS3* might act to enhance suppression. pA, poly A tail.

**Figure 2 ijms-21-08508-f002:**
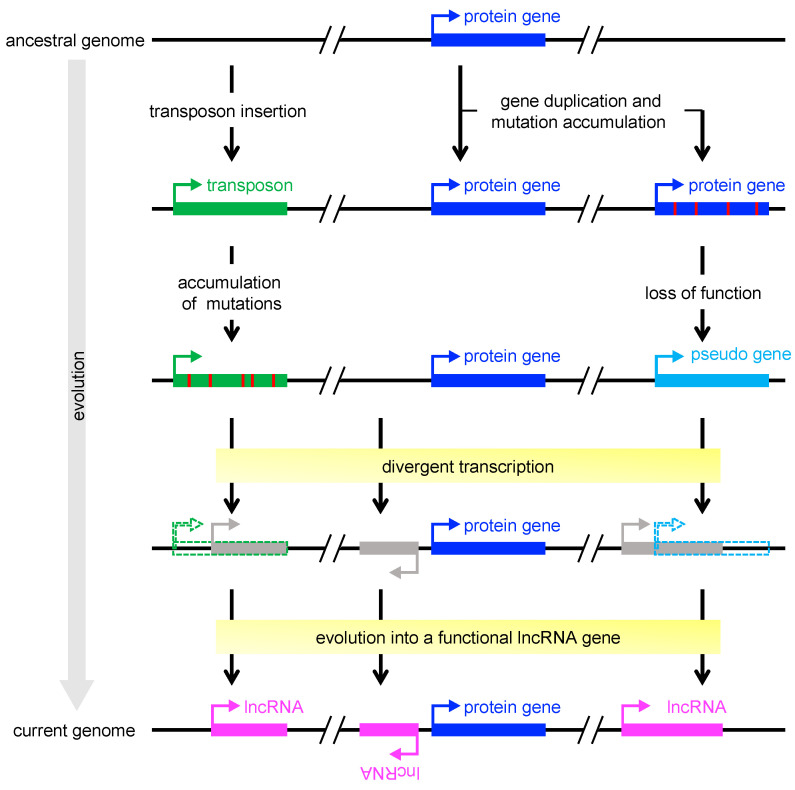
Schematic model of lncRNA emergence. A transcription unit of long non-coding RNA (lncRNA) could originate from pre-existing transcription regulatory sequences of transposable elements (**left**), protein-coding genes (**middle**), or pseudogenes (**right**). Transposable elements could be a source of sequences and signals essential for transcription (e.g., transcription start sites) and processing (e.g., splice and polyadenylation sites) [130], whereas protein-coding genes and pseudogenes could provide transcription factor–binding sites that serve as promoters and enhancers [133]. Gray boxes indicate transcribed loci of unknown function. Red vertical bars indicate mutations.

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
