# Peer review of "Roles of Non-Coding RNAs in Response to Nitrogen Availability in Plants"

_ijms, 2020, doi:10.3390/ijms21228508_

Round 1
Reviewer 1 Report
This short review article nicely summarized the works that have been done on the topic of non-coding RNAs, including miRNAs, siRNAs and long non-coding RNAs, related to nitrogen response and utilization in Arabidopsis and a few other model plant species. An overview of plant non-coding RNAs was also provided on the top of the main topic. I have no problem for the article to be published in the International Journal of Molecular Science but do have a few suggestions.
- A large portion of the “Conclusions and Perspectives” was devoted to the hypotheses of the origin of long non-coding RNAs. From my point of view, it’s inappropriate to introduce the models in this section as a) they are less related to the main topic of the article; b) it’s not a perspective but weakened the section of “conclusions and perspectives”. I suggest to move the evolution models of long non-coding RNAs to section 2.3 (Long regulatory ncRNAs) and also to briefly introduce the current hypotheses for the origin of miRNAs in section 2.2 (small regulatory ncRNAs). Or alternatively, the evolution of lncRNAs and miRNAs can be presented under a separate section.
- Many examples of N-responsive ncRNAs were listed in the article and it was also mentioned in the “Conclusions and Perspectives” that “Engineering of N-responsive ncRNAs could facilitate development of crops with improved N-use efficiency and/or low-N tolerance” and that “ncRNAs are attractive targets of genome engineering …” The authors should elaborate on how ncRNAs can be engineered and describe the researches required for that to happen.
- Suggest to replace “N” at the beginning of Abstract with “Nitrogen”.
Author Response
#Reviewer 1
We greatly appreciate reviewer 1 for constructive suggestions on our manuscript. Responses to each comment are as follows.
Comment 1:
A large portion of the “Conclusions and Perspectives” was devoted to the hypotheses of the origin of long non-coding RNAs. From my point of view, it’s inappropriate to introduce the models in this section as a) they are less related to the main topic of the article; b) it’s not a perspective but weakened the section of “conclusions and perspectives”. I suggest to move the evolution models of long non-coding RNAs to section 2.3 (Long regulatory ncRNAs) and also to briefly introduce the current hypotheses for the origin of miRNAs in section 2.2 (small regulatory ncRNAs). Or alternatively, the evolution of lncRNAs and miRNAs can be presented under a separate section.
Response to the comment:
We agree with the viewpoint of this reviewer and have added the new section "5. Evolution of ncRNAs" to describe the current model of ncRNA evolution (lines 288-314). In this section, we also briefly explained current models of the evolution of plant miRNAs (lines 307-313).
Comment 2:
Many examples of N-responsive ncRNAs were listed in the article and it was also mentioned in the “Conclusions and Perspectives” that “Engineering of N-responsive ncRNAs could facilitate development of crops with improved N-use efficiency and/or low-N tolerance” and that “ncRNAs are attractive targets of genome engineering …” The authors should elaborate on how ncRNAs can be engineered and describe the researches required for that to happen.
Response to the comment:
We have included the following previous report to describe specific examples of engineering ncRNA in the revised version: Yu et al. 2018 showed that overexpressing miR169o in rice promotes N use efficiency (lines 329-330). In addition, we mentioned a potential problem of constitutive overexpression and necessity of development of improved scheme for ncRNA engineering (lines 332-335).
Comment 3:
Suggest to replace “N” at the beginning of Abstract with “Nitrogen”.
Response to the comment:
We revised it as suggested (line 13).

Reviewer 2 Report
Enjoyed reading this article review.
I found it very well written and reach of interesting information highlighting the current knowledge on ncRNAs associated with plant response to N deficiency, particularly miRNAs and long non-coding RNAs.
I have only some concerns/curiosities that I reported following:
l. 229-262 Besides Arabidopsis and monocots, do the authors know whether N-responsive miRNAs (and/or lnc-RNAs) have been identified in herbaceous dicots species, such as tomato? For instance, the work by Pant et al. that the authors mentioned at line 140 (Identification of nutrient responsive Arabidopsis and rapeseed microRNAs by comprehensive real-time PCR profiling and small RNA sequencing. Plant Physiol. 2009, 150, 1541–1555) identified miRNAs involved in nutrient stress in rapeseed as well as Arabidopsis.
l. 138-273 The works described here essentially concerned the identification of N-responsive ncRNAs by application of microarray or high-throughput sequencing technologies. Such studies report useful indications on the miRNAs targeting genes involved in the plant response to low N environment. Nevertheless, results from these researches are largely descriptive, as "omics" analyses alone do not lead to testable hypotheses. Accordingly, they do not provide experimental evidence clearly demonstrating the biological function of miRNAs and/or the causal relationships with other crucial players (e.g. phytohormones) associated with the studied molecular pathways.
Based on the above, I am wondering if a mechanistic link between low N conditions-miRNA regulating networks-hormonal signalling cascades has been proved elsewhere, for instance by analysing knock-out mutants or transgenic plants overexpressing a target miRNA. One example could be a recent survey by Yu et al. (Overexpression of miR169o, an Overlapping MicroRNA in Response to Both Nitrogen Limitation and Bacterial Infection, Promotes Nitrogen Use Efficiency and Susceptibility to Bacterial Blight in Rice. Plant Cell physiol 2018), in which miRNAs induced by low N availability and bacterial infection were first identified by deep sequencing, then a selected candidate, miR169o, was functionally characterised by overexpressing its precursor gene in rice and then subjecting the obtained transgenic plants to stress exposure making comparison with the wild-type.
The above comment could also be useful to widen the discussion reported at lines 275-287 in the Conclusion section.
Minor points:
l.91 For better clarity, I would add (MIR genes) after miRNA host genes; then, I would specify RNA polymerase II
l.97 ..to the target mRNA
l.88-115 I would end the paragraph by mentioning another important subject: the amplification of silencing signals, which occurs via cell-to-cell or systemic delivery of sRNAs, thus briefly highlighting the role of this phenomenon in the regulation of sRNA-mediated responses to stress (details can be found in some recent reviews, such as Pagliarani and Gambino, 2019. Small RNA Mobility: Spread of RNA Silencing Effectors and its Effect on Developmental Processes and Stress Adaptation in Plants. IJMS 20, 4306). In addition, adding this part could be helpful to better introduce the reader to the key role that sRNAs play in the modulation of defence processes and, therefore, to the article section describing N responsive ncRNAs.
l.237: there is a typo here: it should be DISTRIBUTION
Author Response
#Reviewer 2
We deeply appreciate reviewer 2 for the critical comments and suggestions on our manuscript. Response s to each comment are as follows.
Comment 1:
l229-262 Besides Arabidopsis and monocots, do the authors know whether N-responsive miRNAs (and/or lnc-RNAs) have been identified in herbaceous dicots species, such as tomato? For instance, the work by Pant et al. that the authors mentioned at line 140 (Identification of nutrient responsive Arabidopsis and rapeseed microRNAs by comprehensive real-time PCR profiling and small RNA sequencing. Plant Physiol. 2009, 150, 1541–1555) identified miRNAs involved in nutrient stress in rapeseed as well as Arabidopsis.
Response to the comment:
As far as we know, N-responsive ncRNAs have not yet been explored in tomato species, but there are several reports describing N-responsive miRNAs in other crop species including both dicots and monocots that we didn’t mention in the previous manuscript. Therefore, we added brief information regarding this at the end of section 4 (lines 281-286).
Comment 2:
l138-273 The works described here essentially concerned the identification of N-responsive ncRNAs by application of microarray or high-throughput sequencing technologies. Such studies report useful indications on the miRNAs targeting genes involved in the plant response to low N environment. Nevertheless, results from these researches are largely descriptive, as "omics" analyses alone do not lead to testable hypotheses. Accordingly, they do not provide experimental evidence clearly demonstrating the biological function of miRNAs and/or the causal relationships with other crucial players (e.g. phytohormones) associated with the studied molecular pathways. Based on the above, I am wondering if a mechanistic link between low N conditions-miRNA regulating networks-hormonal signaling cascades has been proved elsewhere, for instance by analyzing knock-out mutants or transgenic plants overexpressing a target miRNA. One example could be a recent survey by Yu et al. (Overexpression of miR169o, an Overlapping MicroRNA in Response to Both Nitrogen Limitation and Bacterial Infection, Promotes Nitrogen Use Efficiency and Susceptibility to Bacterial Blight in Rice. Plant Cell physiol 2018), in which miRNAs induced by low N availability and bacterial infection were first identified by deep sequencing, then a selected candidate, miR169o, was functionally characterized by overexpressing its precursor gene in rice and then subjecting the obtained transgenic plants to stress exposure making comparison with the wild-type. The above comment could also be useful to widen the discussion reported at lines 275-287 in the Conclusion section.
Response to the comment:
Thank you for the suggestion with introducing a specific reference. We couldn’t find any other papers reporting N-responsive miRNAs identified by a sort of “omics” analysis with an experimental validation using mutants or transgenic lines. As this reviewer mentioned, a large part of N-responsive ncRNAs identified by “omics” are likely not validated experimentally for its role in N response. As for lncRNAs, our previous study [30] identified TAS3 as the N-responsive lncRNA by a transcriptome analysis and showed the role of TAS3 in N-responsive root morphogenesis by using a transgenic line, which is described in lines 190-205. Some of the N-responsive miRNAs were experimentally proven to regulate genes involved in the hormonal signaling pathway (described in 3.1. miRNAs involved in N nutrition), and omics-identified miRNAs could be involved in plant N response currently unknown. What the reviewer mentioned about the weakness of “omics” is quite true and we think further studies on ncRNAs that provide experimental validation of their biological function are of great importance in the coming decades. Descriptions related to the above have been added in lines 190 and 324-326. Regarding the discussion at lines 275-287 in the previous manuscript, we have added specific examples of “engineering N-responsive ncRNAs” by referring the paper you mentioned to widen the discussion (lines 329-335).
Comment 3:
l.91 For better clarity, I would add (MIR genes) after miRNA host genes; then, I would specify RNA polymerase II
Response to the comment:
We added the word MIR genes after miRNA host genes (lane 91) and specified the word polymerase II as RNA polymerase II (lane 91).
Comment 4:
l.97 ..to the target mRNA
Response to the comment:
We changed the description “the target RNA” to “the target mRNA” (lane 98).
Comment 5:
l.88-115 I would end the paragraph by mentioning another important subject: the amplification of silencing signals, which occurs via cell-to-cell or systemic delivery of sRNAs, thus briefly highlighting the role of this phenomenon in the regulation of sRNA-mediated responses to stress (details can be found in some recent reviews, such as Pagliarani and Gambino, 2019. Small RNA Mobility: Spread of RNA Silencing Effectors and its Effect on Developmental Processes and Stress Adaptation in Plants. IJMS 20, 4306). In addition, adding this part could be helpful to better introduce the reader to the key role that sRNAs play in the modulation of defence processes and, therefore, to the article section describing N responsive ncRNAs.
Response to the comment:
We appreciate the constructive suggestion. Importance of considering small RNA mobility has been described in the end of the section 2.2 “Small regulatory ncRNAs” with including additional reference the reviewer suggested (lines 116-121)
Comment 6:
l.237: there is a typo here: it should be DISTRIBUTION
Response to the comment:
The typo was corrected (lane 243).

Reviewer 3 Report
I am writing to you in regard to manuscript # ijms-991348 entitled "Roles of non-coding RNAs in response to nitrogen availability in plants", which you submitted to Int. J. Mol. Sci. I would like to appreciate the nice research done by authors and well documented in manuscript form. The review is self explanatory and provides an insight to role of non-coding RNA in nitrogen assimilation. I will recommend the acceptance of this article in it current form.
Author Response
#Reviewer 3
We thank to reviewer 3 for the positive review. As attached below, the reviewer does not require any revisions.
Comments and Suggestions for Authors
I am writing to you in regard to manuscript # ijms-991348 entitled "Roles of non-coding RNAs in response to nitrogen availability in plants", which you submitted to Int. J. Mol. Sci. I would like to appreciate the nice research done by authors and well documented in manuscript form. The review is self explanatory and provides an insight to role of non-coding RNA in nitrogen assimilation. I will recommend the acceptance of this article in it current form.
